# Temporal Trend of Severe Obesity in Brazilian State Capitals (2006–2021)

**Flávia Santos Batista Dias** [1] , **Tiago Feitosa da Silva** [1], **Yara de Moura Magalhães Lima** [1], **Luana Silva de Farias** [1], **Jhonatan Gomes Gadelha** [2] and **Alanderson Alves Ramalho** [1,2,*]

1   Postgraduate Program in Public Health, Federal University of Acre, Rio Branco 69920-900, AC, Brazil
2   Postgraduate Program in Health Sciences in Western Amazon, Federal University of Acre, Rio Branco 69920-900, AC, Brazil
*   Correspondence: alanderson.ramalho@ufac.br

**Abstract:** The aim of this study was to analyze the trend of severe obesity in the capitals of the Brazilian states and the Federal District, from 2006 to 2021. For this purpose, a time-series, population-based, observational study was designed using data from the VIGITEL Survey. The dependent variable of this study was the prevalence of severe obesity, which was defined as a body mass index $\geq 40 \ \mathrm{kg/m^2}$. Time series analysis was conducted using Joinpoint Regression Analysis Software v.4.9.1.0. In this study, a normal distribution was assumed, and the average annual percentage change (AAPC) and 95% confidence intervals (95% CIs) were presented. In total, 778,445 individuals participated in the study (38.2% were male and 61.8% were female). The prevalence of severe obesity has increased from 1.1% in 2006 to 1.9% in 2021. The average annual percentage change indicates an upward trend for the period (AAPC: 4.7; 95% CI: 3.8; 5.7). When stratifying the trend of severe obesity by sex, a significant upward trend was observed for females (AAPC: 4.3; 95% CI: 1.9; 6.8). There was a significant upward trend for all age groups, skin colors, and education levels. However, the older age groups and those with less education had lower AAPC.

**Keywords:** obesity; morbid obesity; nutrition surveys; epidemiological surveys; time series studies; noncommunicable diseases; health surveys; health promotion





## 1. Introduction

The World Health Organization (WHO) defines obesity as the excessive accumulation of fat that is harmful to an individual's health [1]. The causes of obesity are plural, including heredity, sociocultural, and economic factors. Obesity is also a risk factor for other non-communicable diseases such as heart disease, diabetes, cancer, and osteoarthritis, among others [2].

The WHO classifies the degree of risk of obesity by the Body Mass Index (BMI), with obesity defined by a BMI $\geq 30 \ \mathrm{kg/m^2}$ and severe obesity by a BMI $\geq 40 \ \mathrm{kg/m^2}$ [3]. Severe obesity increases the risk of morbidity and mortality rates, as it is associated with heart disease, stroke, type 2 diabetes mellitus, metabolic and neoplastic diseases, and orthopedic diseases, among others [2,4]. Poorer outcomes from COVID-19 are also strongly related to severe obesity [5].

This public health problem contributes to functional disability, decreased quality of life, and reduced life expectancy, and has been identified as one of the main causes of death worldwide [6–10]. Studies show that the total number of deaths and disability-adjusted life years (DALYs) attributable to high BMI has increased exponentially in the last three decades (between 1990 and 2017). Globally, in 2017, high BMI contributed to 34.1 million deaths and 1.2 billion DALYs, with significant effects on both sexes, including 70.7 million DALYs in women and 77.0 million DALYs in men [9,11]. Approximately one-fifth of the avoidable disability-adjusted life years (DALYs) associated with non-communicable diseases (NCDs) can be attributed to high body mass index (BMI), which is considered

one of the main risk factors for NCDs [2]. In 2019, the estimated global DALYs and deaths attributed to high BMI represented a significant proportion of all preventable NCDs. The DALYs attributed to obesity are expected to increase by 39.8% over the next 10 years, with a greater increase anticipated in women (41.3%) than in men (38.4%). The prognosis for obesity-related mortality predicts a 42.7% increase from 2020 to 2030, resulting in a global rise from 5,185,364 to 7,397,615 cases, with a greater increase expected in women (43.8%) than in men (41.5%) [2].

Taking into account the economic impact of both direct and indirect costs, Brazil spent approximately US$ 39 billion on overweight and obesity in 2019. Direct costs include medical expenses for individuals with obesity and associated health complications, as well as costs incurred by individuals for transportation to medical appointments and treatment. Indirect costs encompass losses in productivity and labor due to health issues and premature mortality [2,4].

A survey of the health information systems of the Unified Health System (SUS) in Brazil estimated that in 2018, more than US$ 890 million was spent on the treatment of diseases such as hypertension, diabetes, and obesity. Of these expenses, 11% were attributed to obesity, and when considered as a risk factor for hypertension and diabetes, the costs attributable to this disease reached 41% of the total costs [12].

The prevalence of obesity has tripled in the last forty years (between 1975 and 2016). In 2016, 650 million people were obese, which represented about 13% of the world's population (11% of men and 15% of women) [1]. Latin America and the Caribbean are showing increasing obesity trends, with Brazil having one of the highest prevalence [2,13]. The National Health Survey ("Pesquisa Nacional de Saúde"—PNS) in Brazil estimated that the prevalence of obesity (BMI $\geq$ 30 kg/m$^2$) in 2019 was 25.9% (21.8% in men and 29.5% in women) [14]. In Brazilian adults, obesity (BMI $\geq$ 30 kg/m$^2$) has been increasing by approximately 2.0% per year since 2010. Projections suggest that by 2030, 29.7% of Brazilians will be obese (25.87% men and 33.25% women). These same projections suggest that 2.6% of Brazilians will have severe obesity (BMI $\geq$ 40 kg/m$^2$) in 2030 (1.38% men and 3.82% women) [2].

Obesity exponentially increases the risk of morbidity and mortality, especially when the BMI is between 30 and 35 kg/m$^2$. In fact, when the BMI exceeds 35 kg/m$^2$, the risk of premature death doubles [4,15]. A BMI of 35 kg/m$^2$ or higher is associated with an increased risk of premature mortality and decreased quality of life. The comorbidities related to severe obesity represent a significant burden for healthcare systems worldwide. In fact, severe obesity (BMI > 40 kg/m$^2$) has the highest mortality rates globally. In Latin countries, it is estimated that around 200,000 deaths annually are attributed to obesity-related comorbidities. Among men aged 25 to 40 years, the mortality rate is 12 times higher for those with a BMI of 35 kg/m$^2$ or higher compared to individuals within the normal BMI range of 18.5–24.9 kg/m$^2$ [4].

Understanding the trends of severe obesity in Brazil can help in developing goals aimed at reducing the impact of diseases related to severe obesity on the quality of life and life expectancy of affected individuals. Therefore, the objective of this study is to analyze the trend of severe obesity in the capitals of the Brazilian states and the Federal District from 2006 to 2021.

## 2. Materials and Methods

A time-series, population-based, observational study was designed using data from the System for Surveillance of Risk and Protective Factors for Chronic Diseases by Telephone Survey (VIGITEL) from 2006 to 2021. VIGITEL used probabilistic samples of the adult population (18 years or older) residing in the 26 state capitals of Brazil and the Federal District (including the capital Brasília).

The sampling procedures used by the VIGITEL system aimed to obtain probabilistic samples from the population of adults living in households with at least one landline telephone in the year. The system established a minimum sample size of 1500 individuals

aged 18 years or over to estimate the prevalence of risk and protective factors with a 95% confidence interval and a maximum expected error of three percentage points for the estimates. At each active residential landline, an adult resident was randomly selected to participate in the interview. Pregnant women and those who did not know if they were pregnant at the time of the interview were excluded from this study.

The dependent variable of this study was the prevalence of severe obesity, which was defined as a body mass index (BMI) $\geq 40$ kg/m$^2$. BMI was calculated from weight in kilograms divided by the square of height in meters, both self-reported, in response to the questions: "Do you know your weight (even if it is an approximate value)?" and "Sir/Madam, do you know your height?" The independent variables included sex (male; female), age range (18 to 34 years old; 35 to 59 years old; 60 years old or older), skin color (white; non-white), schooling (0 to 8 years; 9 to 11 years; 12 years or more), geographical region (North; Northeast; Midwest; Southeast; South), and Federal District and the capital of the 26 states (Aracaju; Belém; Belo Horizonte; Boa Vista; Campo Grande; Cuiabá; Curitiba; Florianópolis; Fortaleza; Goiânia; João Pessoa; Macapá; Maceió; Manaus; Natal; Palmas; Porto Alegre; Porto Velho; Recife; Rio Branco; Rio de Janeiro; Salvador; São Luís; São Paulo; Teresina; Vitória) (Figure 1).

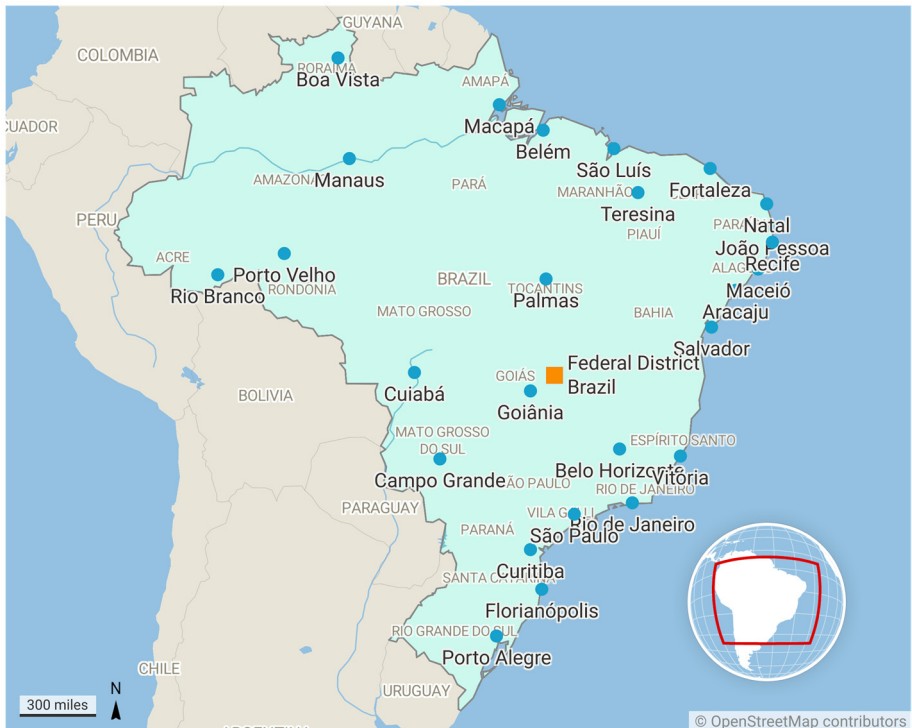

**Figure 1.** Map of Brazil showing the geographical location of the state capitals and the Federal District. The red box indicates the geographical location of Brazil within South America on a global map.

Since landline coverage in Brazilian capitals varies, a post-stratification weight was used for each individual in the final sample of each year to estimate the frequencies of each variable. This weight was calculated using the Rake method [16], which uses interactive procedures to equate the sociodemographic distribution of the VIGITEL sample to the estimated distribution for the total population of the city [17]. The post-stratification weight considered the distribution of sex, age group, and education of the Demographic Census projections and was used to generate all the estimates provided by the system for each capital and the Federal District and for the population as a whole.

Time series analysis was conducted using Joinpoint Regression Analysis Software v.4.9.1.0 (The National Cancer Institute, Bethesda, MD, USA). The software detects linear patterns in the distribution of estimates of interest in two or more time intervals, identifying

segments that have specific trends in estimates through the Annual Percentage Change (APC) and Average Annual Percent Change (AAPC).

The AAPC is a summary measure of the trend over a pre-specified fixed interval. It allows us to use a single number to describe the average APCs over a period of multiple years. It is valid even if the joinpoint model indicates that there were changes in trends during those years. It is computed as a weighted average of the APCs from the joinpoint model, with the weights equal to the length of the APC interval. In this study, a normal distribution was assumed, and the AAPC and 95% confidence intervals were presented in the results, taking into account the value of $p < 0.05$.

Since this research used data of public use and access, made available by Brazilian Ministry of Health in an unrestricted way and without nominal identification, that research waives ethical appreciation in the terms of the National Health Council Resolution (CNS 466/12), which gives provisions for research involving human beings in Brazil.

## 3. Results

From 2006 to 2021, 784,479 individuals aged 18 years or over were interviewed using the VIGITEL survey, 37.9% male and 62.1% female. For this study, 6034 women who were pregnant or who did not know they were pregnant (1.2% of the female population of the study) were excluded. A total of 778,445 individuals participated in this study, of which 38.2% were male and 61.8% were female.

The prevalence of severe obesity has increased from 1.1% in 2006 to 1.9% in 2021. The average annual percentage change indicates an upward trend for the period (AAPC: 4.7; 95% CI: 3.8; 5.7; Table 1).

When stratifying the trend of severe obesity by sex, a significant upward trend was observed for females (AAPC: 4.3; 95% CI: 1.9; 6.8). There was a significant upward trend for all age groups, skin colors, and education levels. However, the older age groups and those with less education had lower AAPC (Table 1).

When stratifying the trend of severe obesity by geographic region, higher AAPCs were observed for the Southeast (AAPC: 4.6; 95% CI: 2.1; 7.2), Northeast (AAPC: 4.4; 95% CI: 3.3; 5.5), and Midwest (AAPC: 4.0; 95% CI: 1.3; 6.9). The North region was the only one that did not show a significant upward trend (Table 1).

In 2021, the capitals with the highest prevalence of severe obesity were Maceió (3.9%), Aracaju (3.8%), Recife (3.2%), Rio Branco (3.1%), Porto Velho (2.7%), Belo Horizonte (2.4%), Natal (2.4%), Porto Alegre (2.3%), Boa Vista (2.2%), João Pessoa (2.2%), and Rio de Janeiro (2.1%) (Table 2).

In the trend analysis stratified by location, the capitals of Belém, Campo Grande, Cuiabá, Florianópolis, Macapá, Manaus, Porto Velho, Salvador, and São Luís did not show a statistically significant trend. The others showed a significant upward trend (Table 2), with differences between the sexes (Tables 3 and 4).

**Table 1.** Temporal trend of severe obesity in the population aged 18 years and over in Brazilian state capitals and the Federal District, according to sociodemographic characteristics. Surveillance of Risk and Protective Factors for Chronic Diseases by Telephone Survey (VIGITEL), 2006 to 2021.

| Variables | 2006 | 2007 | 2008 | 2009 | 2010 | 2011 | 2012 | 2013 | 2014 | 2015 | 2016 | 2017 | 2018 | 2019 | 2020 | 2021 | AAPC | 95% CI | *p*-Value |
|---|---|---|---|---|---|---|---|---|---|---|---|---|---|---|---|---|---|---|---|
| TOTAL | 1.1 | 1.0 | 0.9 | 1.2 | 1.2 | 1.2 | 1.5 | 1.5 | 1.6 | 1.7 | 1.6 | 1.7 | 1.8 | 1.8 | 2.0 | 1.9 | 4.7 | 3.8; 5.7 | **<0.001** |
| **Sex** | | | | | | | | | | | | | | | | | | | |
| Male | 0.9 | 0.8 | 0.5 | 0.7 | 0.8 | 1.1 | 1.0 | 1.2 | 1.2 | 1.1 | 1.2 | 1.4 | 1.4 | 1.3 | 1.6 | 1.6 | 3.5 | −2.7; 10.0 | 0.274 |
| Female | 1.3 | 1.2 | 1.3 | 1.6 | 1.6 | 2.3 | 2.0 | 1.8 | 1.9 | 2.3 | 2.0 | 1.9 | 1.9 | 2.2 | 2.4 | 2.1 | 4.3 | 1.9; 6.8 | **<0.001** |
| **Age group (in years)** | | | | | | | | | | | | | | | | | | | |
| 18–34 | 0.6 | 0.7 | 0.7 | 0.8 | 0.6 | 1.5 | 1.0 | 0.8 | 1.3 | 1.5 | 1.3 | 1.2 | 1.2 | 1.3 | 1.6 | 1.1 | 4.7 | 1.7; 7.8 | **0.004** |
| 35–59 | 1.3 | 1.4 | 1.1 | 1.4 | 1.7 | 2.0 | 1.9 | 1.9 | 1.6 | 2.0 | 2.0 | 2.0 | 2.0 | 2.0 | 2.4 | 2.3 | 3.7 | 2.4; 5.0 | **<0.001** |
| 60 or more | 1.7 | 0.9 | 1.2 | 1.6 | 1.6 | 1.6 | 2.0 | 2.1 | 2.1 | 1.6 | 1.5 | 1.9 | 1.9 | 2.1 | 2.1 | 2.5 | 3.2 | 1.3; 5.3 | **0.003** |
| **Skin color** | | | | | | | | | | | | | | | | | | | |
| White | 1.2 | 0.9 | 0.8 | 1.0 | 0.9 | 1.1 | 1.8 | 1.4 | 1.3 | 1.2 | 1.4 | 1.6 | 1.6 | 1.5 | 1.6 | 1.8 | 3.9 | 1.9; 5.9 | **0.001** |
| Non-white | 1.0 | 1.1 | 1.0 | 1.3 | 1.4 | 1.2 | 1.3 | 1.4 | 1.5 | 1.7 | 1.7 | 1.7 | 1.9 | 1.9 | 2.4 | 1.9 | 5.2 | 4.1; 6.3 | **<0.001** |
| **Level of schooling (by years of studying)** | | | | | | | | | | | | | | | | | | | |
| 0–8 | 1.8 | 1.6 | 1.2 | 1.8 | 2.0 | 2.3 | 2.2 | 2.3 | 2.6 | 2.3 | 2.7 | 2.8 | 2.8 | 2.5 | 3.4 | 2.9 | 4.6 | 3.2; 6.0 | **<0.001** |
| 9–11 | 0.5 | 0.5 | 0.8 | 0.9 | 0.7 | 1.6 | 1.4 | 0.9 | 1.3 | 1.6 | 1.3 | 1.4 | 1.4 | 1.8 | 1.4 | 1.7 | 5.9 | 2.8; 9.1 | **0.001** |
| 12 or more | 0.5 | 0.6 | 0.6 | 0.6 | 0.7 | 1.0 | 0.9 | 1.2 | 0.6 | 1.0 | 1.0 | 0.9 | 0.9 | 1.1 | 1.7 | 1.3 | 6.0 | 3.4; 8.7 | **<0.001** |
| **Geographic region** | | | | | | | | | | | | | | | | | | | |
| North | 1.3 | 1.1 | 1.2 | 1.4 | 1.4 | 2.3 | 1.8 | 1.3 | 1.7 | 2.3 | 1.6 | 1.8 | 1.8 | 1.8 | 1.9 | 1.5 | 2.2 | −0.3; 4.7 | 0.080 |
| Northeast | 0.9 | 1.2 | 1.2 | 1.4 | 1.4 | 1.4 | 1.3 | 1.5 | 1.8 | 1.4 | 1.8 | 1.8 | 1.8 | 1.8 | 2.0 | 2.2 | 4.4 | 3.3; 5.5 | **<0.001** |
| Midwest | 0.6 | 0.8 | 0.9 | 0.8 | 0.8 | 1.0 | 1.3 | 1.2 | 1.9 | 1.0 | 1.3 | 1.2 | 1.2 | 1.2 | 1.3 | 1.5 | 4.0 | 1.3; 6.9 | **0.007** |
| Southeast | 1.3 | 1.0 | 0.6 | 1.1 | 1.2 | 2.0 | 1.7 | 1.7 | 1.4 | 2.0 | 1.7 | 1.8 | 1.8 | 2.0 | 2.4 | 1.9 | 4.6 | 2.1; 7.2 | **0.001** |
| South | 1.1 | 0.8 | 1.2 | 0.9 | 1.1 | 1.5 | 1.3 | 1.3 | 1.5 | 1.5 | 1.4 | 1.5 | 1.5 | 1.4 | 1.2 | 1.9 | 3.2 | 1.4; 5.0 | **0.002** |

**Table 2.** Temporal trend of severe obesity in the population aged 18 years and over in Brazilian capitals and the Federal District, by location. Surveillance of Risk and Protective Factors for Chronic Diseases by Telephone Survey (VIGITEL), 2006 to 2021.

| Capitals | 2006 | 2007 | 2008 | 2009 | 2010 | 2011 | 2012 | 2013 | 2014 | 2015 | 2016 | 2017 | 2018 | 2019 | 2020 | 2021 | AAPC | 95% CI | *p*-Value |
|---|---|---|---|---|---|---|---|---|---|---|---|---|---|---|---|---|---|---|---|
| Aracaju | 1.4 | 1.5 | 1.0 | 1.4 | 2.0 | 1.4 | 2.4 | 1.4 | 1.0 | 1.6 | 2.4 | 2.1 | 1.5 | 1.7 | 2.1 | 3.8 | 4.9 | 1.4; 8.5 | **0.009** |
| Belém | 1.5 | 1.0 | 1.0 | 1.4 | 1.0 | 0.8 | 1.8 | 1.1 | 1.6 | 2.0 | 1.6 | 1.1 | 1.8 | 1.3 | 2.1 | 1.0 | 2.3 | −1.1; 5.7 | 0.169 |
| Belo Horizonte | 0.7 | 1.0 | 1.0 | 1.4 | 1.1 | 0.9 | 1.2 | 1.3 | 1.2 | 1.4 | 1.2 | 1.8 | 1.5 | 1.9 | 1.4 | 2.4 | 5.6 | 3.4; 7.7 | **<0.001** |
| Boa Vista | 0.9 | 1.0 | 1.5 | 0.9 | 1.2 | 0.6 | 1.2 | 1.0 | 1.8 | 1.0 | 0.5 | 1.6 | 1.5 | 2.7 | 1.9 | 2.2 | 5.9 | 2.1; 9.9 | **0.004** |
| Campo Grande | 0.8 | 1.1 | 0.9 | 1.4 | 1.6 | 1.4 | 1.4 | 1.2 | 2.1 | 2.4 | 2.1 | 2.5 | 1.9 | 1.1 | 1.8 | 1.3 | 2.0 | −3.3; 7.6 | 0.47 |
| Cuiabá | 1.2 | 1.1 | 0.7 | 1.1 | 1.5 | 1.9 | 1.6 | 2.4 | 2.9 | 1.0 | 1.3 | 1.6 | 1.8 | 1.7 | 1.1 | 1.5 | 1.6 | −2.9; 6.3 | 0.471 |
| Curitiba | 1.0 | 0.8 | 1.2 | 0.6 | 1.0 | 0.9 | 1.5 | 1.1 | 1.4 | 1.2 | 1.3 | 1.5 | 1.0 | 1.2 | 1.0 | 1.7 | 2.8 | 0.2; 5.4 | **0.038** |
| Florianópolis | 0.6 | 0.9 | 0.8 | 0.9 | 0.8 | 0.8 | 1.7 | 0.8 | 1.9 | 1.6 | 0.9 | 0.8 | 1.3 | 1.2 | 1.5 | 1.3 | 3.6 | −0.4; 7.7 | 0.073 |
| Fortaleza | 0.7 | 0.9 | 1.3 | 1.1 | 1.2 | 1.3 | 0.7 | 1.5 | 2.9 | 1.4 | 1.8 | 1.7 | 1.6 | 1.4 | 1.6 | 1.8 | 4.1 | 0.1; 8.3 | **0.048** |
| Goiânia | 0.7 | 0.8 | 0.6 | 0.8 | 0.9 | 0.7 | 1.0 | 0.9 | 1.0 | 0.8 | 1.6 | 0.9 | 1.7 | 1.2 | 1.4 | 1.3 | 5.4 | 2.9; 8.0 | **<0.001** |
| João Pessoa | 1.1 | 0.8 | 2.0 | 1.4 | 1.2 | 1.5 | 1.3 | 1.5 | 1.1 | 1.1 | 2.0 | 1.6 | 1.6 | 1.2 | 2.5 | 2.2 | 3.5 | 0.5; 6.6 | **0.027** |
| Macapá | 1.0 | 1.5 | 1.2 | 1.7 | 2.8 | 1.4 | 1.6 | 1.3 | 2.1 | 2.2 | 1.9 | 3.4 | 2.0 | 2.9 | 1.9 | 1.0 | 3.3 | −1.0; 7.8 | 0.123 |
| Maceió | 0.5 | 1.3 | 2.2 | 2.8 | 1.0 | 1.6 | 1.0 | 2.2 | 1.3 | 1.9 | 2.4 | 2.2 | 2.0 | 1.6 | 3.8 | 3.9 | 5.7 | 1.3; 10.4 | **0.015** |
| Manaus | 1.4 | 1.1 | 1.1 | 1.5 | 1.4 | 1.6 | 1.9 | 1.5 | 1.7 | 3.2 | 1.6 | 1.7 | 2.0 | 2.0 | 1.7 | 1.2 | 2.5 | −1.0; 6.2 | 0.147 |
| Natal | 1.3 | 1.5 | 0.7 | 1.4 | 1.4 | 1.8 | 1.9 | 1.4 | 2.1 | 2.1 | 1.9 | 1.6 | 1.4 | 3.8 | 2.1 | 2.4 | 5.4 | 2.0; 8.9 | **0.004** |
| Palmas | 0.9 | 0.5 | 0.7 | 0.7 | 1.1 | 0.9 | 1.9 | 1.1 | 1.5 | 0.5 | 1.3 | 1.2 | 0.7 | 0.9 | 1.3 | 3.0 | 6.5 | 1.3; 11.9 | **0.017** |
| Porto Alegre | 1.3 | 0.7 | 1.4 | 1.3 | 1.4 | 1.0 | 0.9 | 1.6 | 1.4 | 1.9 | 1.8 | 1.7 | 2.3 | 1.6 | 1.4 | 2.3 | 4.4 | 1.7; 7.1 | **0.003** |
| Porto Velho | 0.6 | 1.4 | 1.6 | 1.6 | 1.9 | 2.0 | 1.7 | 1.7 | 1.6 | 2.0 | 1.6 | 2.5 | 1.3 | 0.9 | 2.0 | 2.7 | 2.9 | −0.5; 6.4 | 0.085 |
| Recife | 1.0 | 1.2 | 1.2 | 1.6 | 1.6 | 1.3 | 1.7 | 1.8 | 1.5 | 1.5 | 1.9 | 2.0 | 2.2 | 2.5 | 3.1 | 3.2 | 6.9 | 5.3; 8.6 | **<0.001** |
| Rio Branco | 0.9 | 1.5 | 2.1 | 1.5 | 2.1 | 1.8 | 1.5 | 1.1 | 1.9 | 2.5 | 2.4 | 2.7 | 1.9 | 2.6 | 2.5 | 3.1 | 4.8 | 2.2; 7.3 | **0.001** |
| Rio de Janeiro | 1.0 | 1.1 | 1.0 | 1.4 | 1.5 | 1.3 | 1.8 | 1.9 | 1.7 | 1.8 | 2.4 | 1.7 | 2.4 | 1.7 | 1.4 | 2.1 | 4.2 | 1.7; 6.7 | **0.003** |
| Salvador | 0.8 | 1.7 | 1.4 | 1.5 | 1.9 | 1.2 | 1.4 | 1.5 | 1.7 | 1.1 | 1.9 | 1.9 | 1.8 | 1.9 | 1.0 | 2.0 | 1.9 | −0.8; 4.7 | 0.155 |
| São Luís | 0.9 | 1.1 | 0.3 | 1.1 | 1.0 | 1.2 | 1.2 | 1.0 | 1.0 | 0.8 | 0.9 | 1.0 | 1.7 | 0.8 | 1.3 | 1.0 | 1.5 | −1.7; 4.8 | 0.333 |
| São Paulo | 1.6 | 0.9 | 0.3 | 0.9 | 1.0 | 1.0 | 1.8 | 1.6 | 1.2 | 2.2 | 1.5 | 1.8 | 1.7 | 2.1 | 3.3 | 1.7 | 6.3 | 2.3; 10.4 | **0.004** |
| Teresina | 0.6 | 0.8 | 0.5 | 0.6 | 1.2 | 0.8 | 0.6 | 1.4 | 1.7 | 1.3 | 1.2 | 1.8 | 2.6 | 1.3 | 2.3 | 1.0 | 8.3 | 3.6; 13.1 | **0.002** |
| Vitória | 0.8 | 0.6 | 0.7 | 1.1 | 1.3 | 1.2 | 1.0 | 1.2 | 2.0 | 1.4 | 0.6 | 1.6 | 1.7 | 2.3 | 1.6 | 1.9 | 6.3 | 3.2; 9.4 | **0.001** |
| Federal District | 0.4 | 0.6 | 1.0 | 0.7 | 0.4 | 0.9 | 1.3 | 1.0 | 2.1 | 0.6 | 0.9 | 0.8 | 1.1 | 1.2 | 1.2 | 1.6 | 4.5 | 0.4; 8.8 | **0.034** |

**Table 3.** Temporal trend of severe obesity in the male population aged 18 years and over in Brazilian capitals and the Federal District. Surveillance of Risk and Protective Factors for Chronic Diseases by Telephone Survey (VIGITEL), 2006 to 2021.

| Capitals | 2006 | 2007 | 2008 | 2009 | 2010 | 2011 | 2012 | 2013 | 2014 | 2015 | 2016 | 2017 | 2018 | 2019 | 2020 | 2021 | AAPC | 95% CI | *p*-Value |
|---|---|---|---|---|---|---|---|---|---|---|---|---|---|---|---|---|---|---|---|
| Aracaju | 1.3 | 0.5 | 0.4 | 1.1 | 1.9 | 0.9 | 2.2 | 0.9 | 1.2 | 1.1 | 2.1 | 2.8 | 1.4 | 0.5 | 1.8 | 5.2 | 9.7 | 3.2; 16.5 | **0.005** |
| Belém | 1.4 | 1.1 | 0.5 | 0.8 | 1.1 | 0.4 | 0.7 | 0.5 | 0.9 | 1.5 | 0.9 | 0.8 | 2.1 | 0.8 | 2.7 | 0.2 | 5.0 | −1.2; 11.6 | 0.106 |
| Belo Horizonte | 0.4 | 0.9 | 0.5 | 1.1 | 0.6 | 0.6 | 0.7 | 1.1 | 1.1 | 0.5 | 0.3 | 1.6 | 1.2 | 1.7 | 0.7 | 2.5 | 8.2 | 3.1; 13.5 | **0.004** |
| Boa Vista | 0.6 | 0.9 | 1.3 | 0.6 | 0.7 | 0.4 | 0.9 | 0.7 | 2.5 | 0.8 | 0.8 | 1.3 | 1.5 | 3.5 | 1.1 | 1.6 | 7.8 | 1.2; 14.8 | **0.024** |
| Campo Grande | 0.7 | 0.4 | 0.5 | 0.5 | 1.4 | 0.6 | 0.8 | 0.9 | 1.5 | 1.8 | 2.0 | 2.5 | 2.6 | 0.5 | 1.6 | 0.7 | 3.5 | −10.7; 20.0 | 0.650 |
| Cuiabá | 0.6 | 0.6 | 0.7 | 0.7 | 1.1 | 1.3 | 1.0 | 2.1 | 2.4 | 0.5 | 0.4 | 1.2 | 2.0 | 1.3 | 1.4 | 0.2 | 4.2 | −2.9; 11.9 | 0.232 |
| Curitiba | 0.9 | 0.4 | 0.6 | 0.3 | 0.9 | 0.5 | 1.4 | 1.5 | 1.5 | 0.7 | 1.8 | 1.2 | 1.0 | 0.9 | 1.0 | 1.9 | 5.2 | 0.2; 10.5 | **0.042** |
| Florianópolis | 0.2 | 0.9 | 0.9 | 0.4 | 0.6 | 0.7 | 1.6 | 0.6 | 2.8 | 1.5 | 1.3 | 0.7 | 1.8 | 0.5 | 0.8 | 1.3 | 3.6 | −4.2; 12.0 | 0.346 |
| Fortaleza | 0.6 | 0.3 | 0.9 | 0.4 | 1.5 | 0.4 | 0.5 | 1.1 | 2.2 | 1.2 | 1.3 | 1.6 | 1.4 | 0.4 | 1.7 | 1.8 | 6.3 | 0.3; 12.5 | **0.039** |
| Goiânia | 0.4 | 0.6 | 0.4 | 0.7 | 0.6 | 0.5 | 1.3 | 0.6 | 0.2 | 0.3 | 1.9 | 0.6 | 0.9 | 1.1 | 0.9 | 2.0 | 8.0 | 1.9; 14.4 | **0.013** |
| João Pessoa | 1.1 | 0.9 | 1.6 | 0.6 | 0.8 | 1.4 | 1.1 | 1.0 | 1.1 | 0.8 | 2.6 | 0.8 | 0.9 | 0.6 | 4.9 | 2.0 | 8.8 | 2.0; 16.1 | **0.014** |
| Macapá | 0.5 | 0.8 | 0.5 | 0.9 | 0.7 | 0.3 | 0.9 | 1.4 | 1.1 | 2.2 | 1.7 | 5.1 | 0.9 | 2.4 | 1.6 | 0.6 | 6.8 | −5.8; 21.1 | 0.311 |
| Maceió | 0.3 | 0.8 | 1.2 | 2.3 | 0.8 | 1.1 | 0.6 | 0.8 | 0.5 | 1.3 | 1.5 | 1.7 | 1.6 | 0.6 | 2.1 | 3.9 | 7.6 | 1.5; 14.1 | **0.018** |
| Manaus | 0.8 | 1.0 | 0.6 | 0.7 | 1.3 | 1.3 | 0.7 | 1.2 | 1.6 | 2.4 | 1.7 | 1.1 | 1.9 | 1.5 | 1.5 | 1.2 | 4.6 | 0.5; 8.9 | **0.030** |
| Natal | 1.4 | 0.9 | 0.5 | 0.8 | 0.8 | 1.9 | 2.0 | 1.4 | 1.8 | 1.6 | 1.4 | 1.4 | 0.8 | 5.6 | 2.5 | 3.2 | 10.1 | 3.9; 16.5 | **0.003** |
| Palmas | 0.5 | 0.2 | 0.1 | 0.8 | 0.8 | 0.4 | 1.3 | 0.9 | 1.4 | 0.6 | 1.1 | 1.1 | 0.8 | 1.3 | 1.8 | 5.4 | 15.5 | 3.9; 28.3 | **0.007** |
| Porto Alegre | 1.2 | 0.6 | 0.9 | 0.3 | 1.2 | 0.8 | 0.5 | 2.3 | 1.2 | 1.7 | 1.1 | 1.0 | 2.7 | 1.8 | 2.0 | 1.6 | 6.1 | 1.0; 11.5 | **0.021** |
| Porto Velho | 0.2 | 0.9 | 2.0 | 0.9 | 0.8 | 1.0 | 1.2 | 1.8 | 1.7 | 1.2 | 1.7 | 3.3 | 1.0 | 0.7 | 2.1 | 2.2 | 5.2 | −0.6; 11.3 | 0.077 |
| Recife | 0.6 | 0.3 | 0.3 | 2.1 | 0.8 | 1.1 | 0.5 | 1.4 | 0.5 | 1.0 | 1.1 | 2.2 | 0.6 | 2.2 | 2.9 | 0.9 | 6.9 | −0.2; 14.5 | 0.056 |
| Rio Branco | 0.3 | 1.4 | 1.7 | 0.6 | 1.0 | 0.9 | 1.2 | 0.8 | 2.8 | 1.7 | 2.4 | 3.4 | 1.2 | 2.1 | 1.3 | 2.6 | 6.2 | 0.4; 12.4 | **0.038** |
| Rio de Janeiro | 0.9 | 0.9 | 0.4 | 1.0 | 1.0 | 1.2 | 1.2 | 0.6 | 1.0 | 0.4 | 1.7 | 1.3 | 1.4 | 0.9 | 0.2 | 2.4 | 5.3 | 0.3; 10.6 | **0.039** |
| Salvador | 0.4 | 1.6 | 0.4 | 0.9 | 0.9 | 0.8 | 0.6 | 0.5 | 0.5 | 0.6 | 1.2 | 1.5 | 1.3 | 2.1 | 0.5 | 2.8 | 7.3 | 1.3; 13.5 | **0.019** |
| São Luís | 0.9 | 0.2 | 0.1 | 0.8 | 0.5 | 0.7 | 0.9 | 0.5 | 0.2 | 0.6 | 1.0 | 0.7 | 2.0 | 0.7 | 1.0 | 0.8 | 4.8 | −1.4; 11.4 | 0.123 |
| São Paulo | 1.4 | 0.9 | 0.3 | 0.2 | 0.5 | 0.9 | 1.4 | 1.8 | 1.1 | 1.6 | 0.8 | 1.7 | 1.7 | 1.3 | 2.4 | 0.7 | 4.6 | −0.9; 10.3 | 0.097 |
| Teresina | 0.2 | 0.4 | 0.4 | 0.3 | 0.4 | 1.0 | 0.2 | 0.9 | 0.4 | 0.7 | 0.7 | 0.9 | 2.6 | 0.8 | 3.1 | 0.2 | 16.7 | 7.7; 26.4 | **0.001** |
| Vitória | 0.3 | 0.6 | 0.1 | 0.6 | 1.1 | 1.0 | 0.5 | 0.7 | 2.2 | 0.6 | 0.3 | 1.5 | 0.7 | 2.8 | 1.8 | 2.3 | 10.8 | 4.0; 18.1 | **0.004** |
| Federal District | 0.2 | 0.2 | 0.7 | 0.3 | 0.5 | 0.1 | 0.6 | 0.6 | 1.4 | 0.5 | 0.9 | 0.1 | 1.7 | 1.2 | 0.5 | 1.1 | 8.7 | 1.4; 16.6 | **0.022** |

**Table 4.** Temporal trend of severe obesity in the female population aged 18 years and over in Brazilian capitals and the Federal District. Surveillance of Risk and Protective Factors for Chronic Diseases by Telephone Survey (VIGITEL), 2006 to 2021.

| Capitals | 2006 | 2007 | 2008 | 2009 | 2010 | 2011 | 2012 | 2013 | 2014 | 2015 | 2016 | 2017 | 2018 | 2019 | 2020 | 2021 | AAPC | 95% CI | *p*-Value |
|---|---|---|---|---|---|---|---|---|---|---|---|---|---|---|---|---|---|---|---|
| Aracaju | 1.5 | 2.3 | 1.5 | 1.6 | 2.0 | 1.8 | 2.5 | 1.8 | 0.9 | 2.0 | 2.6 | 1.5 | 1.6 | 2.7 | 2.3 | 2.6 | 2.1 | −0.7; 5.0 | 0.127 |
| Belém | 1.7 | 0.9 | 1.4 | 1.9 | 0.9 | 1.1 | 2.8 | 1.6 | 2.3 | 2.4 | 2.3 | 1.3 | 1.6 | 1.7 | 1.6 | 1.8 | 1.3 | −2.5; 5.3 | 0.487 |
| Belo Horizonte | 1.0 | 1.0 | 1.3 | 1.6 | 1.5 | 1.2 | 1.6 | 1.5 | 1.4 | 2.2 | 2.0 | 2.0 | 1.7 | 2.1 | 2.0 | 2.4 | 4.8 | 3.1; 6.5 | **<0.001** |
| Boa Vista | 1.1 | 1.2 | 1.6 | 1.2 | 1.7 | 0.8 | 1.4 | 1.3 | 1.0 | 1.1 | 0.3 | 1.8 | 1.5 | 1.9 | 2.6 | 2.8 | 5.1 | 1.4; 8.9 | **0.010** |
| Campo Grande | 1.0 | 1.8 | 1.3 | 2.2 | 1.8 | 2.1 | 2.0 | 1.4 | 2.7 | 2.9 | 2.1 | 2.4 | 1.3 | 1.6 | 2.0 | 2.0 | 1.5 | −1.8; 4.9 | 0.341 |
| Cuiabá | 1.7 | 1.7 | 0.7 | 1.5 | 1.9 | 2.4 | 2.2 | 2.7 | 3.3 | 1.6 | 2.0 | 2.0 | 1.5 | 2.1 | 0.8 | 2.8 | 1.6 | −2.6; 6.0 | 0.432 |
| Curitiba | 1.1 | 1.1 | 1.6 | 0.9 | 1.0 | 1.3 | 1.6 | 0.8 | 1.4 | 1.6 | 0.8 | 1.9 | 1.1 | 1.4 | 1.0 | 1.5 | 1.1 | −2.0; 4.2 | 0.457 |
| Florianópolis | 0.9 | 0.9 | 0.8 | 1.4 | 1.0 | 0.9 | 1.7 | 0.9 | 1.0 | 1.7 | 0.5 | 0.8 | 0.8 | 1.8 | 2.0 | 1.3 | 3.4 | −0.7; 7.6 | 0.100 |
| Fortaleza | 0.9 | 1.5 | 1.6 | 1.7 | 0.9 | 2.0 | 0.8 | 1.9 | 3.6 | 1.6 | 2.2 | 1.8 | 1.7 | 2.4 | 1.6 | 1.7 | 2.6 | −1.9; 7.3 | 0.239 |
| Goiânia | 1.0 | 1.0 | 0.8 | 0.9 | 1.1 | 0.9 | 0.9 | 1.2 | 1.6 | 1.2 | 1.4 | 1.1 | 2.4 | 1.3 | 1.8 | 0.7 | 4.4 | 0.7; 8.2 | **0.022** |
| João Pessoa | 1.1 | 0.7 | 2.3 | 2.0 | 1.5 | 1.5 | 1.5 | 1.9 | 1.0 | 1.4 | 1.5 | 2.3 | 2.2 | 1.7 | 0.5 | 2.4 | 1.9 | −2.2; 6.1 | 0.349 |
| Macapá | 1.4 | 2.1 | 1.9 | 2.4 | 4.9 | 2.5 | 2.2 | 1.3 | 3.0 | 2.2 | 2.1 | 1.7 | 3.0 | 3.3 | 2.3 | 1.4 | −0.3 | −4.8; 4.4 | 0.896 |
| Maceió | 0.6 | 1.7 | 3.1 | 3.3 | 1.3 | 2.0 | 1.3 | 3.3 | 1.9 | 2.3 | 3.0 | 2.6 | 2.4 | 2.4 | 5.1 | 3.9 | 4.8 | 0.4; 9.3 | **0.033** |
| Manaus | 2.0 | 1.3 | 1.6 | 2.2 | 1.5 | 2.0 | 3 | 1.8 | 1.9 | 4.0 | 1.6 | 2.3 | 2.2 | 2.6 | 1.8 | 1.2 | 1.2 | −3.0; 5.5 | 0.558 |
| Natal | 1.2 | 2.1 | 0.9 | 1.9 | 2.0 | 1.8 | 1.8 | 1.3 | 2.3 | 2.5 | 2.2 | 1.7 | 1.9 | 2.2 | 1.9 | 1.6 | 1.4 | −1.3; 4.2 | 0.295 |
| Palmas | 1.3 | 0.9 | 1.3 | 0.5 | 1.4 | 1.5 | 2.4 | 1.3 | 1.6 | 0.4 | 1.5 | 1.2 | 0.7 | 0.6 | 0.8 | 0.8 | −2.9 | −8.0; 2.5 | 0.262 |
| Porto Alegre | 1.4 | 0.9 | 1.9 | 2.2 | 1.6 | 1.2 | 1.3 | 0.9 | 1.5 | 2.0 | 2.4 | 2.2 | 2.0 | 1.5 | 1.0 | 2.9 | 2.9 | −0.8; 6.8 | 0.115 |
| Porto Velho | 1.0 | 1.9 | 1.2 | 2.4 | 3.0 | 3.1 | 2.3 | 1.6 | 1.5 | 2.9 | 1.5 | 1.7 | 1.6 | 1.2 | 1.9 | 3.2 | 6.9 | −6.5; 22.2 | 0.329 |
| Recife | 1.3 | 2.0 | 1.9 | 1.2 | 2.3 | 1.5 | 2.7 | 2.1 | 2.4 | 1.9 | 2.6 | 1.8 | 3.5 | 2.7 | 3.2 | 5.0 | 6.5 | 3.6; 9.5 | **<0.001** |
| Rio Branco | 1.4 | 1.6 | 2.6 | 2.5 | 3.1 | 2.6 | 1.8 | 1.3 | 1.1 | 3.3 | 2.4 | 2.0 | 2.6 | 3.0 | 3.7 | 3.5 | 3.5 | 0.1; 7.0 | 0.044 |
| Rio de Janeiro | 1.0 | 1.3 | 1.6 | 1.7 | 1.9 | 1.4 | 2.4 | 3.0 | 2.3 | 3.0 | 2.9 | 2.1 | 3.3 | 2.4 | 2.3 | 1.9 | 4.1 | −0.5; 9.0 | 0.082 |
| Salvador | 1.2 | 1.7 | 2.3 | 2.0 | 2.7 | 1.5 | 2.0 | 2.3 | 2.7 | 1.5 | 2.5 | 2.2 | 2.3 | 1.7 | 1.3 | 1.3 | −0.6 | −3.8; 2.8 | 0.712 |
| São Luís | 0.9 | 2.0 | 0.4 | 1.3 | 1.4 | 1.6 | 1.5 | 1.4 | 1.6 | 0.9 | 0.8 | 1.2 | 1.3 | 1.0 | 1.5 | 1.2 | −1.1 | −4.5; 2.5 | 0.526 |
| São Paulo | 1.7 | 1.0 | 0.3 | 1.6 | 1.4 | 1.2 | 2.2 | 1.5 | 1.2 | 2.7 | 2.0 | 1.9 | 1.8 | 2.8 | 4.1 | 2.5 | 6.9 | 2.9; 11.0 | **0.002** |
| Teresina | 1.0 | 1.1 | 0.6 | 1.0 | 1.8 | 0.6 | 0.9 | 1.8 | 2.9 | 1.9 | 1.7 | 2.5 | 2.7 | 1.7 | 1.5 | 1.7 | 5.3 | 0.4; 10.4 | **0.035** |
| Vitória | 1.2 | 0.6 | 1.3 | 1.5 | 1.4 | 1.4 | 1.5 | 1.6 | 1.8 | 2.1 | 1.0 | 1.6 | 2.5 | 1.9 | 1.5 | 1.5 | 3.3 | 0.2; 6.4 | **0.038** |
| Federal District | 0.6 | 0.9 | 1.3 | 1.0 | 0.3 | 1.7 | 2.0 | 1.4 | 2.7 | 0.7 | 0.8 | 1.4 | 0.6 | 1.2 | 1.8 | 2.0 | 3.0 | −2.7; 9.0 | 0.288 |

## 4. Discussion

A significant upward trend was observed in the prevalence of total severe obesity in females and in all age groups, irrespective of education and skin color. When stratified by region, only the North region did not show a significant upward trend (Table 1).

The increasing prevalence of obesity worldwide has already been recognized and described by the World Health Organization (WHO), affecting approximately 650 million people in 2016. This growing trend has also been observed in Latin America and the Caribbean, with Brazil being one of the countries with the highest prevalence of this condition [1,2,18–20]. Between 2006 and 2020, the Surveillance System of Risk and Protective Factors for Chronic Diseases by Telephone Survey (VIGITEL) documented an increase in the prevalence of obesity among men (from 11.4% to 20.3%) and women (from 12.1% to 22.6%) [18].

The more significant increase in female obesity may be related to differences in food consumption patterns, energy expenditure, and gender issues [21,22].

A temporal trend analysis of the prevalence of overweight and obesity in the Brazilian adult population, according to sociodemographic characteristics, between 2006 and 2019 pointed to an increase in the frequency of overweight and obese adults in most of the strata studied, mainly among young people with higher school level, either for overweight or obesity [23]. This scenario has been attributed, mainly in middle-income countries like Brazil, to changes in behavior over the years, such as malnutrition and physical inactivity [19,23–25].

According to the Latin American Study of Health and Nutrition, low-income individuals consume less fresh and/or minimally processed foods compared to those with higher income [13].

Government policies to increase the consumption of healthy foods have been implemented in recent years, through the Ministry of Health, such as mandatory food labeling, agreements with the industry to eliminate trans fats and reduce salt in processed and ultra-processed foods, in addition to the definition and respective recommendations published in the 2014 Food Guide for the Brazilian Population [20,26].

Fernandes et al. [27] emphasized that strategic nutrition education policies should shift their focus from promoting calorie counting as a means to combat non-communicable diseases to considering factors such as ingredients, dietary sources, processing, and cooking methods [27]. Biomedical interventions limited to the biological sphere and the treatment of obesity have proven to be ineffective in reducing its prevalence. Therefore, there is a need to promote the adoption of a balanced, healthy, and sustainable diet through the development of equitable food systems [28].

However, the high frequency of replacing meals with snacks and the high prevalence of consumption of sweets and salt, which still exist, reinforce the importance of advancing in other actions and policies [29]. Passos et al. [30] investigated the relationship between the price of ultra-processed foods and the prevalence of obesity in Brazil, using data from the 2008/09 National Family Budget Survey, which included 55,570 households divided into 550 strata. The authors found an inverse association between the price of ultra-processed foods and the prevalence of overweight and obesity in Brazil. Specifically, for each 1.00% increase in the price of ultra-processed foods, there was an average reduction of 0.33% in the prevalence of overweight and 0.59% in the prevalence of obesity. The authors also suggested that taxing ultra-processed foods could be an effective tool in controlling obesity [30].

Despite the increase in obesity trends occurring throughout Brazil, regional differences are notorious [18–20,25]. Thus, understanding the drivers of these regional differences can help guide more promising intervention strategies [23–25,31].

Given Brazil's socio-economic and cultural diversity, as well as globalization and rapid urbanization, unhealthy eating habits and reduced physical activity have contributed to an increase in non-communicable diseases (NCDs), particularly severe obesity, which is a

risk factor for cardiovascular diseases, type 2 diabetes mellitus, stroke, and several types of cancer. These diseases not only decrease life expectancy but also have detrimental effects on an individual's health and quality of life [4,15,32,33].

The available evidence from Brazil indicates that interventions for obesity have various impacts. This includes economic evaluations of prevention strategies involving physical activity and dietary treatment, as well as drug and/or surgical interventions that have been identified as primary practices in patient care [4,32,33].

Bariatric surgery has proven to be efficient and effective in managing moderate to severe obesity in several national health systems. In 2018, a longitudinal study was conducted at the Hospital das Clínicas of the University of São Paulo, Brazil, to estimate the direct health costs and cost-effectiveness of multiple health outcomes before and after bariatric surgery. After the procedure, the average direct costs for hospitalization (−US$ 2762.22; −23.2%), imaging tests (−US $7.53; −0.8%), and drug treatment (−US$ 175.37; −25.7%) decreased during a six-month period before and after the procedure. However, the total direct costs (US$ 1375.37; +138%), outpatient costs (US$ 0.42; +2.4%), and biochemical assessments (US$ 68.96; +63.4%) increased, as patients require follow-up care after the intervention. The direct cost was US$ 61.68 per kilogram of body weight, and there was a decrease of US$ 164.71 per unit of BMI per patient [32].

However, Watanabe et al. [28] point out that individualized or group outpatient care, as well as surgical interventions such as bariatric surgery and drug interventions, even if implemented on a large scale, would not be sufficient to address the primary underlying factors contributing to the problem of obesity in Brazil [28].

For individuals with severe obesity, the Brazilian Ministry of Health recommends an approach that focuses on nutritional education, physical exercise, psychotherapeutic support, and, if necessary, drug therapy. Another recommendation is that these interventions be implemented in primary healthcare or specialized care settings. If the response to these treatments is inadequate after two years, an evaluation for bariatric surgery is proposed, with pre- and post-surgical follow-up conducted by a multidisciplinary team in the same therapeutic environment [28,34].

While these measures have been proven effective at the individual level, other health interventions with a socio-environmental focus aimed at reducing food deserts through intersectoral coordination appear to strengthen the perspective of ensuring access to adequate, healthy, and sustainable food. These proposals need to be developed through collaboration across workplaces, educational sectors, and stricter regulations on food advertising [28,33].

Given that over 75% of the population relies exclusively on public healthcare services and that severe obesity affects more than a million Brazilians, the challenge faced by specialized medical teams in diagnosing, evaluating, and monitoring this condition can increase the risk of obesity-related complications and result in an increase in healthcare expenses [35].

This study may have possible limitations associated with selection and information bias. The Brazilian population is presently experiencing a period of demographic and socioeconomic transition. As a result, the methods used to generate estimates for trend analyses of population risk factor indicators must more accurately reflect these changes. However, household face-to-face surveys that furnish this data are only conducted once every ten years by the Brazil Demographic Census or in surveys such as PNS, which have a six-year interval between surveys. Consequently, these household-based methods are limited in their utility because they do not reflect annual trends in these indicators. In the case of telephone surveys such as VIGITEL, it is feasible to assess population changes annually because the rake method employs solely the simple frequencies of each variable of the population and the sample. As data collection was carried out via a telephone survey using landlines, it is possible that in capitals with low landline coverage, estimates are more imprecise. Although VIGITEL's annual surveys utilized a consistent methodology throughout the observed period, there were fluctuations in fixed telephony coverage in Brazil during that time. As a result, this variability may alter population estimates.

However, VIGITEL uses post-stratification weights to overcome this limitation, and there is evidence that even in cities with low fixed telephone coverage, VIGITEL data are valid [36]. The collection of self-reported data may reflect inaccuracies in the calculation of BMI. However, the methodology used by VIGITEL is widely used and recommended in health surveys [23,37], in addition to being validated for Brazil [37].

This study presents an up-to-date analysis of severe obesity trends in Brazilian state capitals and the Federal District, which may prove useful for monitoring this condition in Brazil and comparing it with other countries that conduct annual surveys with similar methodologies. Further research is required to comprehend regional disparities, related factors, and challenges in treatment management.

## 5. Conclusions

The prevalence of severe obesity increased from 1.1% in 2006 to 1.9% in 2021. The average annual percentage change was 4.7%. When stratifying this trend by sociodemographic characteristics, a significant upward trend was observed for females, all age groups, skin colors and levels of education. Regional disparities were observed. This study can complement others, contributing to a better monitoring of obesity and severe obesity.

**Author Contributions:** Conceptualization, A.A.R.; methodology, A.A.R. and F.S.B.D.; formal analysis, T.F.d.S. and A.A.R.; writing—original draft preparation, F.S.B.D., Y.d.M.M.L. and A.A.R.; writing—review and editing, T.F.d.S., L.S.d.F. and J.G.G.; funding acquisition, A.A.R. All authors have read and agreed to the published version of the manuscript.

**Funding:** This research was funded by the Brazilian Ministry of Health (MoH) as well as the National Council for Scientific and Technological Development (CNPq) (grant number 439701/2018-0) by Public Notice CNPq/MS/SAS/DAB/CGAN No. 26/2018.

**Institutional Review Board Statement:** Ethical review and approval for this study were waived since it used data of public use and access, made available by the Ministry of Health of Brazil in an unrestricted manner and without nominal identifications, under the terms of the National Health Council Resolution, CNS 466/12, which provides for research involving human beings in Brazil.

**Informed Consent Statement:** Informed consent was obtained via the VIGITEL survey from all subjects involved in the study.

**Data Availability Statement:** Data presented in this study are publicly and unrestrictedly available by the Brazilian Ministry of Health on the website of the IT Department of Unified Health System. https://datasus.saude.gov.br/ (accessed on 10 January 2023).

**Conflicts of Interest:** The authors declare no conflict of interest.

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
