# Peer review of "Temporal Trend of Severe Obesity in Brazilian State Capitals (2006–2021)"

_2673-4168, doi:10.3390/obesities3020010_

Round 1

Reviewer 1 Report

Thank you for submitting your article on the trend of severe obesity in the Brazilian capitals and the Federal District from 2006 to 2021. The study design and methodology are well-described, and the results provide valuable information on the increasing prevalence of severe obesity in this population. However, there are some potential pitfalls that should be addressed to improve the clarity and quality of the content.  Firstly, more information about the significance and implications of the study findings can benifit the study. For example, what are the potential health consequences of the increasing prevalence of severe obesity in this population? Are there any implications for public health policy or interventions to address this issue?  Secondly, it would be helpful to include more information about the study limitations and potential sources of bias. For example, were there any limitations to the VIGITEL Survey data that may have affected the accuracy of the prevalence estimates? Were there any factors that may have influenced the trends observed over time?  Thirdly, the abstract could benefit from more specific information about the demographic characteristics of the study population, such as age, gender, and socioeconomic status. This information would help the reader understand the generalizability of the study findings to other populations and settings.  Finally, it may be helpful to include a brief discussion of the potential implications of the study findings for future research in this area. For example, are there any unanswered questions or areas for further investigation that arise from the study results?  Overall, the abstract provides valuable information on the increasing prevalence of severe obesity in the Brazilian capitals and the Federal District. However, addressing the aforementioned pitfalls will help to improve the clarity and quality of the content. 

Author Response

Point 1: Thank you for submitting your article on the trend of severe obesity in the Brazilian capitals and the Federal District from 2006 to 2021. The study design and methodology are well-described, and the results provide valuable information on the increasing prevalence of severe obesity in this population. However, there are some potential pitfalls that should be addressed to improve the clarity and quality of the content.  Firstly, more information about the significance and implications of the study findings can benifit the study. For example, what are the potential health consequences of the increasing prevalence of severe obesity in this population? Are there any implications for public health policy or interventions to address this issue?

Response 1: Thanks for the valuable considerations. As requested, we added to the discussion more information about the significance and implications of the study findings can benefit the study (lines 234 to 276):

Given Brazil's socio-economic and cultural diversity, as well as globalization and rapid urbanization, unhealthy eating habits and reduced physical activity have contributed to an increase in non-communicable diseases (NCDs), particularly severe obesity, which is a risk factor for cardiovascular diseases, type 2 diabetes mellitus, stroke, and several types of cancer. These diseases not only decrease life expectancy but also have detrimental effects on an individual's health and quality of life [4, 15, 32, 33].

The available evidence from Brazil indicates that interventions for obesity have various impacts. This includes economic evaluations of prevention strategies involving physical activity and dietary treatment, as well as drug and/or surgical interventions that have been identified as primary practices in patient care. [4,32,33].

Bariatric surgery has proven to be efficient and effective in managing moderate to severe obesity in several national health systems. In 2018, a longitudinal study was conducted at the Hospital das Clínicas of the University of São Paulo, Brazil, to estimate the direct health costs and cost-effectiveness of multiple health outcomes before and after bariatric surgery. After the procedure, the average direct costs for hospitalization (-US$2,762.22; -23.2%), imaging tests (-US$7.53; -0.8%), and drug treatment (-US$175.37; -25.7%) decreased during a six-month period before and after the procedure. However, the total direct costs (US$1,375.37; +138%), outpatient costs (US$0.42; +2.4%), and biochemical assessments (US$68.96; +63.4%) increased, as patients require follow-up care after the intervention. The direct cost was US$61.68 per kilogram of body weight, and there was a decrease of US$164.71 per unit of BMI per patient [32].

However, Watanabe et al. [28] point out that individualized or group outpatient care, as well as surgical interventions such as bariatric surgery and drug interventions, even if implemented on a large scale, would not be sufficient to address the primary underlying factors contributing to the problem of obesity in Brazil [28].

For individuals with severe obesity, the Brazilian Ministry of Health recommends an approach that focuses on nutritional education, physical exercise, psychotherapeutic support, and, if necessary, drug therapy. Another recommendation is that these interventions be implemented in primary healthcare or specialized care settings. If the response to these treatments is inadequate after two years, an evaluation for bariatric surgery is proposed, with pre- and post-surgical follow-up conducted by a multidisciplinary team in the same therapeutic environment [28, 34].

While these measures have been proven effective at the individual level, other health interventions with a socio-environmental focus aimed at reducing food deserts through intersectoral coordination appear to strengthen the perspective of ensuring access to adequate, healthy, and sustainable food. These proposals need to be developed through collaboration across workplaces, educational sectors, and stricter regulations on food advertising [28,33].

Given that over 3/4 of the population relies exclusively on public healthcare services and that severe obesity affects more than a million Brazilians, the challenge faced by specialized medical teams in diagnosing, evaluating, and monitoring this condition can increase the risk of obesity-related complications and result in an increase in healthcare expenses [35].

Point 2: Secondly, it would be helpful to include more information about the study limitations and potential sources of bias. For example, were there any limitations to the VIGITEL Survey data that may have affected the accuracy of the prevalence estimates? Were there any factors that may have influenced the trends observed over time?

Response 2: We added more information about the study limitations and potential sources of bias (lines 277 to 296):

This study may have possible limitations associated with selection and information bias. The Brazilian population is presently experiencing a period of demographic and socioeconomic transition. As a result, the methods used to generate estimates for trend analyses of population risk factor indicators must more accurately reflect these changes. However, household surveys that furnish this data are only conducted once every ten years by the Brazil Demographic Census or in surveys such as PNS, which have a six-year interval between surveys. Consequently, these household-based methods are limited in their utility because they do not reflect annual trends in these indicators. In the case of telephone surveys like VIGITEL, it is feasible to assess population changes annually because the rake method employs solely the simple frequencies of each variable of the population and the sample. As data collection was carried out by a telephone survey of landlines, it is possible that in capitals with low landline coverage, estimates are more imprecise. Although VIGITEL's annual surveys utilized a consistent methodology throughout the observed period, there were fluctuations in fixed telephony coverage in Brazil during that time. As a result, this variability may alter population estimates. However, Vigitel uses post-stratification weights to overcome this limitation, and there is evidence that even in cities with low fixed telephone coverage, Vigitel data are valid [36]. The collection of self-reported data may reflect inaccuracies in the calculation of BMI. However, the methodology used by VIGITEL is widely used and recommended in health surveys [23, 37], in addition to being validated for Brazil [37].

Point 3: Thirdly, the abstract could benefit from more specific information about the demographic characteristics of the study population, such as age, gender, and socioeconomic status. This information would help the reader understand the generalizability of the study findings to other populations and settings. 

Response 3: We added more information about the demographic characteristics (lines 16 and 17).

Point 4: Finally, it may be helpful to include a brief discussion of the potential implications of the study findings for future research in this area. For example, are there any unanswered questions or areas for further investigation that arise from the study results?  Overall, the abstract provides valuable information on the increasing prevalence of severe obesity in the Brazilian capitals and the Federal District. However, addressing the aforementioned pitfalls will help to improve the clarity and quality of the content. 

Response 4: We added brief discussion of the potential implications of the study findings for future research in this area (lines 297 to 301):

This study presents an up to date analysis of severe obesity trends in Brazilian state capitals and the Federal District, which may prove useful for monitoring this condition in Brazil and comparing it with other countries that conduct annual surveys with similar methodologies. Further research is required to comprehend regional disparities, related factors, and challenges in treatment management.

Reviewer 2 Report

1. Introduction: 

- You have mentioned DALY's for Obesity. But you have broken down gender-specific DALYs for overweight. Do we need to know about the overweight DALY's and do we need it broken down if you haven't done that for obesity? 

- Any DALY data on severe obesity since that's the focus of your topic ?

- The $890 million for Rx of HTN, DM, and obesity seems significantly less compared to the USA, where the cost of obesity and related diseases was $1.3 TRILLION[not billion] dollars. Do you want to get up-to-date data on obesity-related conditions per year?

- What is brazils obesity prevalence? please mention it after global prevalence so readers get some context! 

- Please rearrange some sentences. ' in the period between 2006-2020, the prevalence of obesity in Brazilian capitals and Fed District increased from 11.8%- 21.5%" should be in the last paragraph, and the 2nd last sentence. 

- " who classifies the degree of risk of obesity........... are also strongly related to severe obesity" should come earlier in the introduction as you want to talk about what is obesity and the BMI classification of obesity and severe obesity. 

- Also please mention most common diseases associated with Obesity in order, Heart disease, diabetes, cancers, osteoarthritis etc.

- Please focus introduction of obesity and then severe obesity since thats the main topic. 

- Please talk about the dangers of SEVERE Obesity compared to obesity and why you are doing this study and its significance. 

2. Materials and Methods: 

- please change JOINTPOINT to joinpoint in 1 of the paragraphs. 

4. Discussion: 

- when you are mentioning the upward trend, please inform the readers which table are you talking about in brackets. 

- the 2nd paragraph is repetitive in the Introduction and discussion. Please change it from at least 1 place. 

- " government policies paragraph" - What actions and policies are you talking about. please elaborate. Please revise the entire paragraph. Not sounding crisp. 

- No specific comments made on the causes on increase in severe obesity- just non specifics mentioned. please elaborate. 

- What are the 'drivers' and the 'interventions strategies' that you mention in the discussion ? I see only non specific sentences and less firm answers and future directions. 

- Telephone surveys do carry a lot of bias. I am afraid very few studies are done to validate their reliability. so there are some limitations to this study as they are not the most representative studies. 

Author Response

Point 1: 1. Introduction: 

- You have mentioned DALY's for Obesity. But you have broken down gender-specific DALYs for overweight. Do we need to know about the overweight DALY's and do we need it broken down if you haven't done that for obesity? 
- Any DALY data on severe obesity since that's the focus of your topic ?

Response 1: Thanks for the valuable considerations. The latest data we found about DALY and overweight and obesity did not stratify by level of severity. Data are grouped under "high BMI". Therefore, we revised the introduction to inform these data, as follows (lines 38 to 53):

This public health problem contributes to functional disability, decreased quality of life, reduced life expectancy, and has been identified as one of the main causes of death worldwide [6-10]. Studies show that the total number of deaths and disability-adjusted life years (DALYs) attributable to high BMI has increased exponentially in the last three decades (between 1990 and 2017). Globally, in 2017, high BMI contributed to 34.1 million deaths and 1.2 billion DALYs, with significant effects on both sexes, including 70.7 million DALYs in women and 77.0 million DALYs in men [9, 11]. Approximately one-fifth of the avoidable disability-adjusted life years (DALYs) associated with non-communicable diseases (NCDs) can be attributed to high body mass index (BMI), which is considered one of the main risk factors for NCDs [2]. In 2019, the estimated global DALYs and deaths attributed to high BMI represented a significant proportion of all preventable NCDs. The DALYs attributed to obesity are expected to increase by 39.8% over the next 10 years, with a greater increase anticipated in women (41.3%) than in men (38.4%). The prognosis for obesity-related mortality predicts a 42.7% increase from 2020 to 2030, resulting in a global rise from 5,185,364 to 7,397,615 cases, with a greater increase expected in women (43.8%) than in men (41.5%) [2].

Point 2: - The $890 million for Rx of HTN, DM, and obesity seems significantly less compared to the USA, where the cost of obesity and related diseases was $1.3 TRILLION[not billion] dollars. Do you want to get up-to-date data on obesity-related conditions per year?

Response 2: We changed the text to be clearer and added the latest available data (lines 54 to 64):

Taking into account the economic impact of both direct and indirect costs, Brazil spent approximately US$ 39 billion on overweight and obesity in 2019. Direct costs include medical expenses for individuals with obesity and associated health complications, as well as costs incurred by individuals for transportation to medical appointments and treatment. Indirect costs encompass losses in productivity and labor due to health issues and premature mortality [2, 4].

A survey of the health information systems of the Unified Health System (SUS) in Brazil estimated that in 2018, more than US$890 million was spent on the treatment of diseases such as hypertension, diabetes, and obesity. Of these expenses, 11% were attributed to obesity, and when considered as a risk factor for hypertension and diabetes, the costs attributable to this disease reached 41% of the total costs [12].

Point 3: - What is brazils obesity prevalence? please mention it after global prevalence so readers get some context!

Response 3: We added the information (lines 69 to 75):

The National Health Survey ("Pesquisa Nacional de Saúde" - PNS) in Brazil estimated that the prevalence of obesity (BMI ≥ 30 kg/m²) in 2019 was 25.9% (21.8% in men and 29.5% in women) [14]. In Brazilian adults, obesity (BMI ≥ 30 kg/m2) has been increasing by approximately 2.0% per year since 2010. Projections suggest that by 2030, 29.7% of Brazilians will be obese (25.87% men and 33 .25% women). These same projections suggest that 2.6% of Brazilians will have severe obesity (BMI ≥40 kg/m2) in 2030 (1.38% men and 3.82% women) [2].

Point 4: - Please rearrange some sentences. ' in the period between 2006-2020, the prevalence of obesity in Brazilian capitals and Fed District increased from 11.8%- 21.5%" should be in the last paragraph, and the 2nd last sentence. AND  " who classifies the degree of risk of obesity........... are also strongly related to severe obesity" should come earlier in the introduction as you want to talk about what is obesity and the BMI classification of obesity and severe obesity.  Also please mention most common diseases associated with Obesity in order, Heart disease, diabetes, cancers, osteoarthritis etc.  Please focus introduction of obesity and then severe obesity since thats the main topic.  Please talk about the dangers of SEVERE Obesity compared to obesity and why you are doing this study and its significance. 

Response 4: We reorganized the sentences and rewrote the introduction to focus on obesity and severe obesity according to the suggested.

Point 5:  Materials and Methods: - please change JOINTPOINT to joinpoint in 1 of the paragraphs.

Response 5:  Changed to Joinpoint (line 125).

Point 6:  Discussion: - when you are mentioning the upward trend, please inform the readers which table are you talking about in brackets. 

Response 6:  The information was inserted: (Table 1).

Point 7:  - the 2nd paragraph is repetitive in the Introduction and discussion. Please change it from at least 1 place. 

Response 7:  The second paragraph of the discussion has been changed (lines 188 to 195):

The increasing prevalence of obesity worldwide has already been recognized and described by the World Health Organization (WHO), affecting approximately 650 million people in 2016. This growing trend has also been observed in Latin America and the Caribbean, with Brazil being one of the countries with the highest prevalence of this condition [1, 2, 18, 19, 20]. Between 2006 and 2020, the Surveillance System of Risk and Protective Factors for Chronic Diseases by Telephone Survey (VIGITEL) documented an increase in the prevalence of obesity among men (from 11.4% to 20.3%) and women (from 12.1% to 22.6%) [18].

Point 8:  - " government policies paragraph" - What actions and policies are you talking about. please elaborate. Please revise the entire paragraph. Not sounding crisp. No specific comments made on the causes on increase in severe obesity- just non specifics mentioned. please elaborate. What are the 'drivers' and the 'interventions strategies' that you mention in the discussion ? I see only non specific sentences and less firm answers and future directions. 

Response 8: The discussion of the manuscript was extensively revised to cover the possible causes of the increase in severe obesity, current public policies in Brazil and strategies of the Brazilian health system in the management of obesity and severe obesity. We've also added new references to corroborate this discussion.

Point 9:  Telephone surveys do carry a lot of bias. I am afraid very few studies are done to validate their reliability. so there are some limitations to this study as they are not the most representative studies. 

Response 9: We agree that telephone surveys carry biases and limitations. However, national face-to-face household surveys in Brazil do not take place annually. Intervals between surveys of this nature are greater than five years. In this way, the telephone survey becomes useful, even with its limitations, as it is carried out annually and uses the same collection methods. There are validation studies of this survey in Brazil. Therefore, to make the limitations and possible biases clearer, we detail in the manuscript (lines 278 to 296):

The Brazilian population is presently experiencing a period of demographic and socioeconomic transition. As a result, the methods used to generate estimates for trend analyses of population risk factor indicators must more accurately reflect these changes. However, household face-to-face surveys that furnish this data are only conducted once every ten years by the Brazil Demographic Census or in surveys such as PNS, which have a six-year interval between surveys. Consequently, these household-based methods are limited in their utility because they do not reflect annual trends in these indicators. In the case of telephone surveys like VIGITEL, it is feasible to assess population changes annually because the rake method employs solely the simple frequencies of each variable of the population and the sample. As data collection was carried out by a telephone survey of landlines, it is possible that in capitals with low landline coverage, estimates are more imprecise. Although VIGITEL's annual surveys utilized a consistent methodology throughout the observed period, there were fluctuations in fixed telephony coverage in Brazil during that time. As a result, this variability may alter population estimates. However, Vigitel uses post-stratification weights to overcome this limitation, and there is evidence that even in cities with low fixed telephone coverage, Vigitel data are valid [36]. The collection of self-reported data may reflect inaccuracies in the calculation of BMI. However, the methodology used by VIGITEL is widely used and recommended in health surveys [23, 37], in addition to being validated for Brazil [37].

Round 2

Reviewer 2 Report

Thanks for incorporating the suggestions to this paper. Now it looks good. 

The introduction, material, and methods are much more precise. 

The discussion now seems too long. Please consider reducing some sentences which are not high yield, and keep the sentences that are very important to your topic in the paper. try to keep 5-7 sentences in 1 paragraph of the discussion. A paragraph shouldn't be too big or too small. 

Otherwise looks good and ready to go !